# Detection of novel coronaviruses in bats in Myanmar

**Marc T. Valitutto** [1]*, **Ohnmar Aung**[1], **Kyaw Yan Naing Tun**[1], **Megan E. Vodzak**[1¤], **Dawn Zimmerman**[1], **Jennifer H. Yu** [1], **Ye Tun Win**[2], **Min Thein Maw**[2], **Wai Zin Thein**[2], **Htay Htay Win**[2], **Jasjeet Dhanota**[3], **Victoria Ontiveros**[3], **Brett Smith**[3], **Alexandre Tremeau-Brevard**[3], **Tracey Goldstein**[3], **Christine K. Johnson**[3], **Suzan Murray**[1], **Jonna Mazet**[3]

1 Global Health Program, Smithsonian's National Zoological Park and Conservation Biology Institute, Washington, District of Columbia, United States of America, 2 Livestock Breeding and Veterinary Department, Ministry of Agriculture, Livestock and Irrigation, Naypyitaw, Myanmar, 3 One Health Institute, School of Veterinary Medicine, University of California, Davis, California, United States of America

¤ Current address: Vanderbilt Vaccine Center, Vanderbilt University Medical Center, Nashville, Tennessee, United States of America

* ValituttoM@si.edu

**Data Availability Statement:** All sequences are available from the GenBank database. The accession numbers have been included in a supporting document.

## Abstract

The recent emergence of bat-borne zoonotic viruses warrants vigilant surveillance in their natural hosts. Of particular concern is the family of coronaviruses, which includes the causative agents of severe acute respiratory syndrome (SARS), Middle East respiratory syndrome (MERS), and most recently, Coronavirus Disease 2019 (COVID-19), an epidemic of acute respiratory illness originating from Wuhan, China in December 2019. Viral detection, discovery, and surveillance activities were undertaken in Myanmar to identify viruses in animals at high risk contact interfaces with people. Free-ranging bats were captured, and rectal and oral swabs and guano samples collected for coronaviral screening using broadly reactive consensus conventional polymerase chain reaction. Sequences from positives were compared to known coronaviruses. Three novel alphacoronaviruses, three novel betacoronaviruses, and one known alphacoronavirus previously identified in other southeast Asian countries were detected for the first time in bats in Myanmar. Ongoing land use change remains a prominent driver of zoonotic disease emergence in Myanmar, bringing humans into ever closer contact with wildlife, and justifying continued surveillance and vigilance at broad scales.

## Introduction

Infectious diseases are considered to be "emerging" if they appear in a new population or geographic region or are occurring with greater frequency than the expected background rate [1–3]. Emerging infectious diseases (EIDs) are capable of causing debilitating health effects and financial instability, especially in less developed countries with insufficient capacity to mount health interventions, and thus pose a significant global public health challenge in the 21st

**Funding:** This study was made possible by the generous support of the American people through the United States Agency for International Development (USAID) Emerging Pandemic Threats PREDICT project (cooperative agreement number AID-OAA-A-14-00102 and GHN-A-OO-09-00010-00 to JM). https://www.usaid.gov/news-information/fact-sheets/emerging-pandemic-threats-program The contents are the responsibility of the authors and do not necessarily reflect the views of USAID or the United States Government. The sponsor did not play any role in the study design, data collection and analysis, decision to publish, or preparation of the manuscript. Support for the preparation of this manuscript was provided by the Morris Animal Foundation and Dennis and Connie Keller through a training partnership, as well as Judy and John W. McCarter, Jr., and James and Jamie Coss. This content has not been reviewed or endorsed by the Morris Animal Foundation, and the views expressed herein do not necessarily reflect the views of the Foundation, its officers, directors, affiliates, or agents.

**Competing interests:** The authors have declared that no competing interests exist.

century. Jones et al. reported a consistent growth in reported EID events from 1940 to 2004, demonstrating their increasing presence on the global stage [4].

An estimated 60–75% of EIDs are comprised of zoonotic diseases; of these, more than 70% have purportedly originated in wildlife species [3–5]. Spillover has been largely attributed to changes in anthropogenic activity subsequent to exponential human population growth since the latter half of the 20th century. Large-scale land use change, such as deforestation and land conversion for agriculture, can alter host-pathogen relationships and increase human encounter rates with wildlife and their pathogens, making cross-species transmission events more likely [6,7]. For established pathogens, human-mediated biodiversity loss often leads to reduced populations of suboptimal host species and increased numbers of competent or amplifying hosts, potentially precipitating higher infection rates in people [8]. In addition, intensification of livestock and poultry production systems results in artificially dense populations of domestic animals, which can lead to pathogen amplification and spillover to humans [7]. Approximately two-thirds of human pathogens occupy complex, multi-host systems, and pathogens with multiple animal hosts, including some wildlife species, are more likely to become emergent [9].

Bats are increasingly recognized as the natural reservoirs of viruses of public health concern [10–13]. The capacity of bats to carry and transmit zoonotic pathogens has been hypothesized to be due to their unique life history traits, including their ability for sustained flight, potential for long-distance dispersal, aggregation into densely populous colonies, and adaptation to peri-urban habitats [11,12]. Historically, bats have been linked to highly pathogenic viruses that pose a serious threat to human health, including the coronaviruses responsible for severe acute respiratory syndrome (SARS) and Middle East respiratory syndrome (MERS), the hemorrhagic ebola and Marburg filoviruses, and paramyxoviruses such as Nipah virus [10,11,13–18]. More recently, a pandemic of an acute respiratory syndrome originating in Wuhan, China in December 2019 was linked to a coronavirus (designated "SARS-CoV-2") that shared 96% identity with a bat-borne coronavirus at the whole-genome level [19]. In some cases, these viruses can subsequently spread through person-to-person contact following spillover from animals, increasing their epidemic potential [10,11,19].

The 2002–2003 SARS epidemic, the emergence of MERS in people in 2012, and the ongoing COVID-19 pandemic have prompted substantial interest in detecting coronaviruses of bat origin due to public health concern and their pandemic potential [10,13–18]. Coronaviruses (CoV) are a family of enveloped, single-stranded RNA viruses that commonly infect the respiratory and gastrointestinal tracts of their mammalian and avian hosts [10]. The *Alphacoronaviruses* and *Betacoronaviruses* are of particular importance to human health, with SARS-CoV, SARS-CoV-2, and MERS-CoV–which have caused the most severe disease in humans to date–belonging to the latter group [10,20,21]. Mounting evidence indicates that bats are the evolutionary hosts and origin for these CoV lineages [10,19–22]. In addition to human-associated CoVs, bats are also hosts of coronaviruses that infect production animals, and have been implicated in the emergence and origin of swine acute diarrhea syndrome (SADS), transmissible gastroenteritis virus (TGEV) in pigs, and porcine epidemic diarrhea (PED), which can cause considerable losses [23–26]. Thus, bat-borne CoVs can pose a significant threat to human health and food production.

In spite of these infectious disease threats, bats are an indisputably essential component of ecosystems. They provide critical services such as seed dispersal, pollination, control of insect populations (including crop pests and disease vectors), and fertilization via guano, making them invaluable assets to agricultural industries and small-holder farming [27]. The importance of bats to ecosystems and human communities while being the natural reservoirs of many zoonotic pathogens presents a challenge for disease control. The potential threats posed

by bat-borne coronaviruses to human and livestock health necessitate the identification and characterization of these viruses at high-risk interfaces among humans, domestic animals, and wildlife.

Particular attention is needed in developing regions of high biodiversity, where EIDs are most likely to arise, and where substantial losses in agricultural production may be a source of financial insecurity [28–32]. Myanmar is a particularly vulnerable country due to the interplay of ecological and human factors, which increase opportunities for viral spillover. The nation is situated in the heart of the Southeast Asia region, a hotspot for EIDs, including some neglected tropical diseases and some of pandemic potential like SARS and H5N1 influenza [31,32]. A combination of biological, ecological, socioeconomic, and anthropogenic factors renders the region particularly susceptible to emerging zoonoses that could impart a considerable public health and economic burden [31,32]. Our study aimed to detect coronaviruses in free-ranging bats living in close proximity to human communities.

## Materials and methods

### Sampling sites

Between May 2016 and August 2018, sample and data collection occurred at three selected sites in Myanmar: 1) Northern District in Yangon Region, near Hlawga National Park (spanning lat: 17.04˚N, 17.50˚N; long: 95.86˚E, 96.12˚E); 2) Hpa-An in Kayin state (spanning lat: 16.66˚N, 16.88˚N; long: 97.58˚E, 97.68˚E); and 3) Shwebo of Sagaing region (lat: 22.37˚N, long: 95.78˚E) (Fig 1). These sites were targeted as potential high-risk human-animal interfaces due to land use change increasing human proximity to wildlife and potential human exposures through livelihood, recreational, commercial, and religious or cultural activities. Two of these sites also featured popular cave systems where people were routinely exposed to bats through guano harvesting, religious practices, and ecotourism. Sites 1 and 2 consisted of several smaller sub-sites where bat capture and sampling events occurred. All surveillance activities were conducted in collaboration with three of Myanmar's government ministries: (1) the Ministry of Livestock, Agriculture, and Irrigation; (2) the Ministry of Health and Sports; and (3) the Ministry of Natural Resources and Environmental Conservation. All work conducted was approved through a Letter of Agreement, Ethical Review Committee, and Memorandum of Understanding, respectively.

### Animal capture and sampling

Bat sampling was performed by trained field personnel in collaboration with Myanmar's Ministry of Agriculture, Livestock and Irrigation (MOALI) and Ministry of Natural Resources and Environmental Conservation (MONREC). All bats were captured using mist nets, with each individual manually restrained for species identification, morphometric evaluation, and sample collection. No anesthetic or immobilization agents were used during capture or handling. Oral and rectal swabs were collected when possible using sterile polyester-tipped applicators (animal size often precluded rectal swab collection). Naturally voided guano samples consisting of combined urine and feces were also collected from the environment using plastic tarps. At Site 2, the tarps were placed on the floor of the caves and left overnight, with sample collection occurring the following morning. At Sites 1 and 3, the tarps were placed at cave entrances and under roosting areas in the evening as the bats emerged to forage, and samples were collected immediately. Guano pellets were collected randomly from the tarps and pooled. Tarps were disinfected between each use and gloves were changed in between each sampling event. Pooled guano samples were attributed to a presumptive host species based on field identification of species in caves when possible, otherwise were designated as "Unidentified

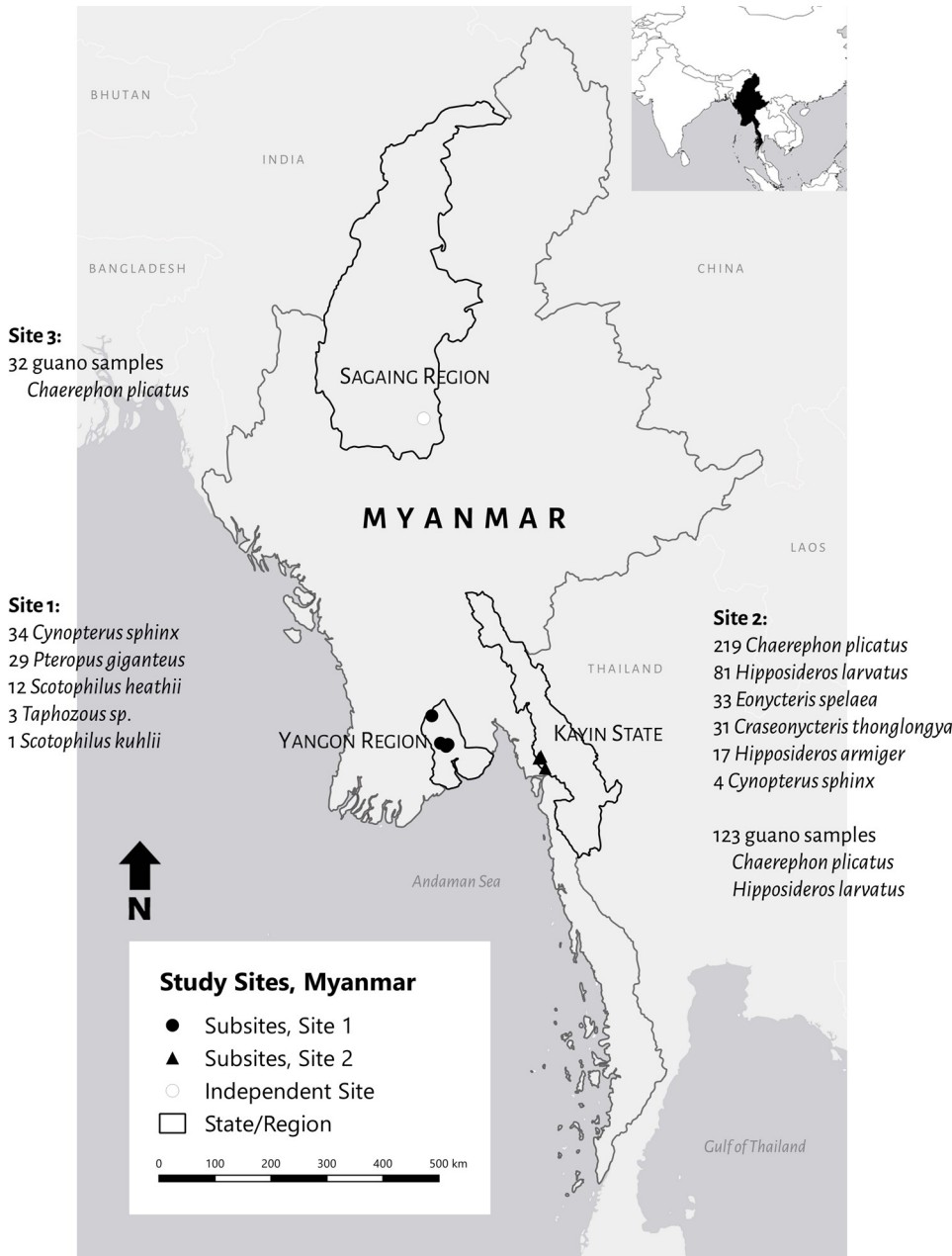

**Fig 1. Myanmar study sites.** Map of bat capture sites in Myanmar, 2016–2018. Data Sources: Natural Earth. Map created in QGIS 2.18.4. 2020.

Chiropterans" when multiple species were present. All sample types were collected into 500 μL viral transport medium (ThermoScientific MicroTest tubes, Fisher Scientific, Pittsburgh, PA, USA) or 500 μL TRIzol reagent (Invitrogen TRIzol reagent, Fisher Scientific, Pittsburgh, PA, USA), transported from the field in liquid nitrogen, and transferred to a -80°C freezer within five days and stored until time of testing. Bats were humanely trapped, handled, and sampled from according to protocols approved by the Institutional Animal Care and Use Committee of the University of California at Davis (Protocol 19300) and Smithsonian Institution (Protocol 16–05) and with approvals from MOALI and MONREC. Bats were released within 1 km of the

capture site as soon as possible upon completion of each sampling event, with net capture and pillowcase restraint between 5 to 30 minutes and handling times less than 5 minutes for each individual.

### RNA extraction, viral detection, and sequencing

Sample testing was performed at the UC Davis One Health Institute Laboratory and the Veterinary Diagnostic Laboratory, Livestock, Breeding, and Veterinary Department (LBVD) in Myanmar. 250 μl was used from each sample for RNA extraction per kit instructions, and to ensure availability of an additional aliquot should a second extraction or other downstream analyses be needed. RNA was extracted using Direct-Zol RNA columns (Zymo Research Corp), and 8 μl RNA was used for cDNA transcription using Superscript III (Invitrogen). Samples were screened for coronaviruses using two broadly reactive consensus conventional polymerase chain reaction (PCR) assays targeting two non-overlapping fragments (434 bp and 332 bp) of the RNA-dependent RNA polymerase (RdRp) of orf1ab of CoVs [33,34]. Bands of the expected size were cloned (pCR4-TOPO vector; Invitrogen Corp.) and Sanger sequenced (ABI 3730 Capillary Electrophoresis Genetic Analyzer; Applied Biosystems, Inc., Foster City, CA).

Sequences were analyzed and edited using Geneious Prime (Version 2019.1.3), uploaded to Genbank (S1 Table), and compared with known sequences in the database. Coronavirus sequences were classified as belonging to viral taxa according to established cut-offs and methods [28]. Virus sequences that shared less than 90% identity to a known sequence were labelled sequentially as PREDICT_CoV-1, -2, -3 etc; while groups sharing ≥90% identity to a sequence already in GenBank were given the same name as the matching sequence. Based on these criteria, the CoV sequences detected were assigned to discrete viral taxa. Viral culture and isolation were not attempted for any positive samples.

### Host DNA barcoding

Bat samples positive for a CoV–including positive pooled guano samples–were barcoded to confirm the host species using PCR assays targeting fragments of the cytochrome B gene (cytB) and the cytochrome oxidase subunit 1 genes (CO1) [35]. One PCR amplicon was selected for sequencing and compared to reference sequences in GenBank using BLAST tools. A threshold of 97% sequence identity was used to confirm the species. Sequences with <95% sequence identity were classified to the genus. DNA barcoding was also performed on a subset of the CoV-negative pooled guano samples. Pooled guano samples were assigned a presumptive origin species based on host barcoding.

## Results

A total of 464 bats representing at least 11 species across eight genera from six families were captured and sampled (Table 1). Both insectivorous microbats and fruit bats were represented in our study population. A total of 759 samples were collected and tested (464 oral swabs, 140 rectal swabs, 155 guano samples). A total of 461 samples were collected in the dry-season sampling (244 oral swabs, 117 rectal swabs, and 100 guano samples) and 298 samples (220 oral swabs, 23 rectal swabs, and 55 guano samples) in the wet season.

CoVs were detected in 48 samples: one oral swab and seven rectal swabs from seven individual bats and 40 pooled guano samples (Table 1). Viral fragments were detected from one unidentified tomb bat (*Taphozous* sp.), three Horsfield's leaf-nosed bats (*Hipposideros larvatus*), and three greater Asiatic yellow house bats (*Scotophilus heathii*). Thirty-six of the 40 positives detected in guano were attributed to *H. larvatus*, while the host species for the remaining four positive pooled guano samples was identified as wrinkle-lipped free-tailed bats

**Table 1. Summary of positives and coronaviruses detected in bats in Myanmar.**

| Taxonomic Level | Common Name | Individual Bats | Rectal Swab | Oral Swab | Pooled Guano | Samples | CoVs Detected |
|---|---|---|---|---|---|---|---|
| | | Pos/Total | Pos/Total | Pos/Total | Pos/Total | Pos/Total | |
| Vespertilioniformes | | | | | | | |
| Vespertilionidae | | | | | | | |
| *Scotophilus heathii* | Greater Asiatic yellow house bat | 3/12[1] | 3/12 | 0/12 | 0/0 | 3/24 | PREDICT_CoV-35[2], 90[3] |
| *Scotophilus kuhlii* | Lesser Asiatic yellow house bat | 0/1 | 0/1 | 0/1 | 0/0 | 0/2 | |
| Emballonuridae | | | | | | | |
| *Taphozous* sp.[4] | | 1/3 | 1/3 | 0/3 | 0/0 | 1/6 | PREDICT_CoV-35[2] |
| Molossidae | | | | | | | |
| *Chaerephon plicatus* | Wrinkle-lipped free-tailed bat | 0/219 | 0/65 | 0/219 | 4/105 | 4/389 | PREDICT_CoV-47,82 |
| Pteropodiformes | | | | | | | |
| Hipposideridae | | | | | | | |
| *Hipposideros armiger* | Great Himalayan leaf-nosed bat | 0/17 | 0/0 | 0/17 | 0/0 | 0/17 | |
| *Hipposideros larvatus* | Horsfield's leaf-nosed bat | 3/81 | 3/16 | 1/81 | 36/50 | 40/147 | PREDICT_CoV-92,93,96[3] |
| Craseonycteridae | | | | | | | |
| *Craseonycteris thonglongyai* | Kitti's hog-nosed bat | 0/31 | 0/9 | 0/31 | 0/0 | 0/40 | |
| Pteropodidae | | | | | | | |
| *Eonycteris spelaea* | Lesser dawn bat | 0/33 | 0/0 | 0/33 | 0/0 | 0/33 | |
| *Cynopterus sphinx* | Greater short-nosed fruit bat | 0/38 | 0/5 | 0/38 | 0/0 | 0/43 | |
| *Pteropus giganteus* | Indian flying fox | 0/29 | 0/29 | 0/29 | 0/0 | 0/58 | |
| Total | | 7/464 | 7/140 | 1/464 | 40/155 | 48/759 | |

[1]Includes *Scotophilus cf. heathii* based on 95–97% shared nt identity with reference sequences.

[2]Virus previously discovered during PREDICT-1 surveillance activities.

[3]Indicates at least one instance of co-infection.

[4]Did not meet the 95% nt identity threshold for identification to the taxonomic level of species.

(*Chaerephon plicatus*). Overall viral prevalence across all bat taxa and all coronaviral genotypes was approximately 1.5%. The vast majority of positive detections (83.3%) were made from pooled guano samples, while oral swabs had the lowest yields. Positive detections were made from 40 samples collected during the dry season (83.3%), while wet-season sampling resulted in positive detections from eight samples (16.7%). Both Sites 1 and 2 accounted for positive detections, while no coronaviral sequences were detected at Site 3.

Fifty-four total sequences were recovered, clustering within seven distinct coronaviral genotypes. Using established cut-offs and methods [28], we detected four alpha coronaviruses (PREDICT_CoV-35, 47, 82, and 90) and three betacoronavirues (PREDICT CoV-92, 93, and 96). Of these, the alphacoronavirus PREDICT_CoV-35 was previously known, having been found in *Scotophilus kuhlii*, unidentified *Myotis*, and other unspeciated host bats in the neighboring countries of Cambodia and Vietnam from 2013 to 2017 [36]. The remaining six coronaviruses were novel (three alphacoronaviruses and three betacoronaviruses). PREDICT_CoV-92 was the most commonly detected coronavirus, found in 36 pooled guano samples attributed to *H. larvatus* (Table 1). Interestingly, three coronaviruses were only found as co-infections: PREDICT_CoV-90 was detected with PREDICT_CoV-35, PREDICT_CoV-93 with -96, and PREDICT_CoV-96 also with -92.

## Discussion

Three new alphacoronaviruses, three new betacoronaviruses, and one previously described alphacoronavirus were detected in bats in Myanmar. None of the viruses appeared to be closely related to SARS-CoV, MERS-CoV, or SARS-CoV-2. Guano samples accounted for the majority of positives, suggestive of an important transmission route for CoV shedding from bats [29,28,29] and a possible risk to people during the act of guano harvesting [37,38]. Viral detection in guano also has implications for future surveillance, as our study demonstrates the value of non-invasive collection of guano for viral surveillance, potentially obviating the need for handling individual bats for coronaviral detection. Our findings supplement those of He et al., who profiled the virome of insectivorous bats from northern Myanmar but did not detect coronaviruses in that study [40].

A difference was found in positives for CoV by species, as samples from *H. larvatus* represented 83% of positives. A wide diversity of CoVs has been found in Hipposiderid bats [28,34,39,41], and our study is consistent with those findings. Four CoVs detected in our surveillance study were found in a single host species each: PREDICT_CoV-90 was found only in *S. heathii*; and PREDICT_CoV-92, -93, and -96 were found only in *H. larvatus* (Table 1). These findings may possibly suggest limited host-switching and viral sharing for certain viruses within our study populations, a pattern consistent with prior observations that viral groups are likely significantly associated with host taxa at the family level [28]. However, further evidence is needed to elucidate host-viral relationships and ecology in the region.

Our findings also likely reflect a bias in our sampling effort. Although *H. larvatus* samples accounted for the most positives, these were largely detected in guano samples collected from the environment, as individuals were not frequently caught by mist net. Overall in our study, the numbers of individual bats handled and sampled per species were relatively low, ranging from one to 218 (Table 1). Viral prevalence may vary widely with the species of host and pathogen. Anthony et al. suggested a sample size of at least 154 individuals per species in order to maximize our ability to detect CoVs. Targeting more host species, specific taxa (Hipposideridae), and larger sample sizes might have improved our detection rate in the species where no CoVs were found [28,29].

Currently, active pathogen surveillance at human-wildlife interfaces in Myanmar is limited. Despite relatively small sample sizes, our study detected several coronaviruses in insectivorous bats, suggesting that more may remain to be uncovered. Given the potential consequences for public health in light of expanding human activity, continued surveillance for coronaviruses is warranted, especially in other species and human-wildlife interfaces. Anthony et al. estimated that over 3,200 CoVs occur in bats, most of which remain undiscovered [28]. Enhancing our sampling effort to incorporate more diverse bat families and larger sample sizes may enable us to identify more CoVs in bats in Myanmar. Additionally, because only short fragments of the conserved RdRp gene (328 bp and 434 bp) were amplified in this study, protein sequence and phylogenetic analyses were not pursued, and identification of recombination events was not possible. While this is an inherent shortcoming of our methodology, the purpose of this study was not to fully characterize specific viruses, but to broadly screen for viruses in bats living in proximity to human communities to better understand potential sources of zoonotic transmission in the context of these human-wildlife interfaces. Further studies may consider complete genomic sequencing for more comprehensive profiling of the bat viromes in this ecosystem. In particular, evaluation of the spike gene sequences may provide insights into host range, including potential viral host-sharing or host-switching events [42].

Land use change will likely continue bringing people into closer proximity with bats, raising encounter rates and opportunities for spillover, facilitating the emergence of zoonotic viruses,

and supporting the need for surveillance [12,43]. Historically, human activities have arguably played a significant role in interspecies transmission events. Following the SARS outbreak, coronaviruses have since been detected in numerous bat species globally, including in Asia, Africa, Europe, the Americas, and the Australasian region [28,44–49]. Mounting evidence supports the role of bats in the transmission of viruses of public health concern–including SARS-CoV and MERS-CoV–and the zoonotic potential of unknown bat-borne coronaviruses warrants vigilant, continued surveillance [10]. Understanding their ecology and prevalence in their natural hosts can improve our ability to detect, prevent, and respond to potential public health threats. Finally, given the essential ecosystem services provided by bats, public health efforts should advocate for preventative measures to protect people against disease transmission while enabling human communities and bats to coexist on a shared landscape.

## Supporting information

**S1 Table. Final edited sequences and genbank accession numbers.**
(XLSX)

## Acknowledgments

We also thank the Livestock Breeding and Veterinary Department (LBVD) within the Ministry of Agriculture, Livestock, and Irrigation (MOALI); Ministry of Natural Resources and Environmental Conservation (MONREC); and the Department of Medical Research (DMR) within the Ministry of Health and Sports (MOHS), Myanmar, with whom we collaborated closely on surveillance activities. Thanks also to the invaluable field and laboratory staff who provided technical skill and expertise and were critical in the research process.

## Author Contributions

**Conceptualization:** Tracey Goldstein, Christine K. Johnson, Jonna Mazet.

**Data curation:** Ohnmar Aung, Tracey Goldstein, Christine K. Johnson, Jonna Mazet.

**Formal analysis:** Marc T. Valitutto, Jennifer H. Yu, Min Thein Maw, Wai Zin Thein, Jasjeet Dhanota, Victoria Ontiveros, Brett Smith, Alexandre Tremeau-Brevard, Tracey Goldstein, Christine K. Johnson.

**Funding acquisition:** Tracey Goldstein, Christine K. Johnson, Suzan Murray, Jonna Mazet.

**Investigation:** Marc T. Valitutto, Ohnmar Aung, Kyaw Yan Naing Tun, Megan E. Vodzak, Dawn Zimmerman, Min Thein Maw, Wai Zin Thein, Htay Htay Win, Jasjeet Dhanota, Victoria Ontiveros, Brett Smith, Alexandre Tremeau-Brevard, Tracey Goldstein, Jonna Mazet.

**Methodology:** Tracey Goldstein, Christine K. Johnson, Jonna Mazet.

**Project administration:** Marc T. Valitutto, Ohnmar Aung, Kyaw Yan Naing Tun, Megan E. Vodzak, Dawn Zimmerman, Ye Tun Win, Min Thein Maw, Tracey Goldstein, Christine K. Johnson, Suzan Murray, Jonna Mazet.

**Resources:** Marc T. Valitutto, Ohnmar Aung, Kyaw Yan Naing Tun, Megan E. Vodzak, Dawn Zimmerman, Ye Tun Win, Min Thein Maw, Htay Htay Win, Jasjeet Dhanota, Victoria Ontiveros, Brett Smith, Alexandre Tremeau-Brevard, Tracey Goldstein, Suzan Murray, Jonna Mazet.

**Software:** Jonna Mazet.

**Supervision:** Marc T. Valitutto, Ohnmar Aung, Kyaw Yan Naing Tun, Megan E. Vodzak, Dawn Zimmerman, Ye Tun Win, Min Thein Maw, Wai Zin Thein, Tracey Goldstein, Christine K. Johnson, Suzan Murray, Jonna Mazet.

**Validation:** Min Thein Maw, Wai Zin Thein, Htay Htay Win, Jasjeet Dhanota, Victoria Onti-veros, Brett Smith, Alexandre Tremeau-Brevard, Tracey Goldstein.

**Visualization:** Marc T. Valitutto.

**Writing – original draft:** Marc T. Valitutto, Jennifer H. Yu.

**Writing – review & editing:** Marc T. Valitutto, Ohnmar Aung, Jennifer H. Yu, Ye Tun Win, Min Thein Maw, Tracey Goldstein, Christine K. Johnson, Suzan Murray, Jonna Mazet.

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
