## [Decision Letter · Decision Letter 0]

7 Jan 2020

PONE-D-19-34833

Detection of Novel Coronaviruses in Bats in Myanmar

PLOS ONE

Dear Dr. Valitutto,

Thank you for submitting your manuscript to PLOS ONE. After careful consideration, we feel that it has merit but does not fully meet PLOS ONE’s publication criteria as it currently stands. Therefore, we invite you to submit a revised version of the manuscript that addresses the points raised during the review process.

This is an important paper to document the diversity of coronavirus in bat in Myanmar. The reviewers agree that this information is important to the field and we wish you to provide extra information as requested by the two reviewers in order to proceed with the manuscript acceptance. 

We would appreciate receiving your revised manuscript by Feb 21 2020 11:59PM. To enhance the reproducibility of your results, we recommend that if applicable you deposit your laboratory protocols in protocols.io, where a protocol can be assigned its own identifier (DOI) such that it can be cited independently in the future. For instructions see: http://journals.plos.org/plosone/s/submission-guidelines#loc-laboratory-protocols

We look forward to receiving your revised manuscript.

Kind regards,

Renee W.Y. Chan, Ph.D.

Academic Editor

PLOS ONE

Journal Requirements:

2. In your Methods section, please provide additional location information of the study sites, including geographic coordinates for the data set if available.

4. We note that you are reporting an analysis of a microarray, next-generation sequencing, or deep sequencing data set. PLOS requires that authors comply with field-specific standards for preparation, recording, and deposition of data in repositories appropriate to their field. Please upload these data to a stable, public repository (such as ArrayExpress, Gene Expression Omnibus (GEO), DNA Data Bank of Japan (DDBJ), NCBI GenBank, NCBI Sequence Read Archive, or EMBL Nucleotide Sequence Database (ENA)). In your revised cover letter, please provide the relevant accession numbers that may be used to access these data. For a full list of recommended repositories, see http://journals.plos.org/plosone/s/data-availability#loc-omics or http://journals.plos.org/plosone/s/data-availability#loc-sequencing.

5. We note that Figure 1 in your submission contains map images which may be copyrighted. All PLOS content is published under the Creative Commons Attribution License (CC BY 4.0), which means that the manuscript, images, and Supporting Information files will be freely available online, and any third party is permitted to access, download, copy, distribute, and use these materials in any way, even commercially, with proper attribution. For these reasons, we cannot publish previously copyrighted maps or satellite images created using proprietary data, such as Google software (Google Maps, Street View, and Earth). For more information, see our copyright guidelines: http://journals.plos.org/plosone/s/licenses-and-copyright.

b).    You may seek permission from the original copyright holder of Figure 1 to publish the content specifically under the CC BY 4.0 license.

b).    If you are unable to obtain permission from the original copyright holder to publish these figures under the CC BY 4.0 license or if the copyright holder’s requirements are incompatible with the CC BY 4.0 license, please either i) remove the figure or ii) supply a replacement figure that complies with the CC BY 4.0 license. Please check copyright information on all replacement figures and update the figure caption with source information. If applicable, please specify in the figure caption text when a figure is similar but not identical to the original image and is therefore for illustrative purposes only.

Reviewers' comments:

Reviewer's Responses to Questions

**Comments to the Author**

1. Is the manuscript technically sound, and do the data support the conclusions?

Reviewer #1: Yes

Reviewer #2: Yes

2. Has the statistical analysis been performed appropriately and rigorously? 

Reviewer #1: Yes

Reviewer #2: N/A

3. Have the authors made all data underlying the findings in their manuscript fully available?

Reviewer #1: Yes

Reviewer #2: Yes

4. Is the manuscript presented in an intelligible fashion and written in standard English?

Reviewer #1: Yes

Reviewer #2: Yes

5. Review Comments to the Author

Reviewer #1: I have uploaded my comments on the manuscript for the author to review and take into consideration when re-writing the manuscript. importantly the phylogenetic tree to show the relationship between the corona viruses.

Reviewer #2: Studies of “Bat borne” coronaviruses are vital for the effective mitigation and prevention of zoonotic coronavirus outbreaks. It is likely that the currently circulating alphacoronaviruses and betacoronaviruses in mammals have their evolutionary ancestral viruses originated from different bat species. Meanwhile most recent coronaviruses that cause human infections like the MERS-CoV or future viruses could possibly have their ancestral relatives in bats. With the ever-expanding human activity and habitat that continue to overlap habitats of bats there is bound to be future coronavirus spillover and subsequent outbreaks at some point. Therefore, we need to urgently invest to conduct long-term coronavirus surveillance studies in bats as well as in other wildlife and livestock to maintain our vigilance. In this context, the authors here report about their viral surveillance attempts in Myanmar to identify viruses that would pose a risk for potential spill over into human population. Despite the modest samples size the authors identify three new alphacoronaviruses and three new betacoronaviruses in bats in Myanmar that warrants further explorations on this. They found many of these positives in guano samples indicating an important transmission route. Overall, I believe this is an important work that needs to be published. However, this report could to be further improved and my comments are as below.

In the Methods section,

it is not clear how much of the initial sample volume used for RNA extraction from each of the oral, rectal and guano samples. Although one would expect, this is specially important since the oral swabs yielded the lowest.

Additionally, if deposited in GenBank or elsewhere the sequence data accession numbers need to be provided and mentioned here.

In the Results and discussion,

Authors need to indicate if at all any virus culture been attempted and if any of these viruses has been isolated and characterized.

The absence of at least RdRp partial sequence phylogenetic analysis is a shortcoming for this report that needs to be discussed or addressed. On a similar note, a phylogenetic analysis derived from spike(S) and receptor-binding domain (RBD) would be informative to include here.

Moreover, this report could be further improved if they consider providing information on the protein sequence alignment of coronavirus RdRps or S proteins with the conserved motifs and other unique signatures indicated as necessary.

Although their limited sequencing results may not allow, if the novel viruses found in this study would constitute recent recombination events from existing coronaviruses needs to be at least discussed since such recombination is rather common and are thought to contribute to the emergence of novel coronaviruses.

6. PLOS authors have the option to publish the peer review history of their article (what does this mean?). If published, this will include your full peer review and any attached files.

Reviewer #1: No

Reviewer #2: No

---

## [Author Response · Author response to Decision Letter 0]

27 Feb 2020

RESPONSES TO REVIEWERS (Comments to the Author)

Reviewer #1: 

I have uploaded my comments on the manuscript for the author to review and take into consideration when re-writing the manuscript. Importantly the phylogenetic tree to show the relationship between the corona viruses.

• Line 65 needs literature quotation

• Lines 68-70 need literature quotation

• Lines 90-92 need literature quotation

Thank you for finding these errors. The manuscript has been revised to include literature citations for these lines listed. 

• Line 110 Animal capture and sampling: General comment is to clarify to the reader if there was any form of anesthesia application on the big chiropterans that couldn’t be handled manually during sampling. 

Thank you for identifying this omission. All animals were handled manually; no anesthetic or immobilization agents were used to assist with capture or restraint, and the manuscript has been corrected to reflect this.

• Also Line 131-132: after how long were the bats released after capture and sampling. Time estimate here would be good for the readers.

Thank you for the suggestion. We have noted in the manuscript the following text: “Bats were released within 1 km of the capture site as soon as possible upon completion of each sampling event, with net capture and pillowcase restraint between 5 to 30 minutes and handling times less than 5 minutes for each individual.” 

• Line 162 Results general comment is mainly on seasonality. Can the season/months be indicated in the table of results as it is stated at length in the results section. 

Thank you for this very reasonable suggestion. However, the authors note that seasonality was intentionally discussed at greater length in the results of the main body of the paper, as the authors did not find a way to naturally insert the data into Table 1, given that the species captured and sites spanned both dry- and wet-seasons. Results not easily depicted in the table were discussed at greater length in the main body. 

• Still in the results section; - a phylogenetic tree is needed to indicate to the reader the relationship between the beta and alpha corona viruses with the known and unknown corona viruses.

Thank you for the suggestion. However, the authors note that only short fragments of the RdRp gene (328bp and 434 bp, depending on the assay used) were amplified. Given the short sequence fragment of a conserved gene, phylogenetic analyses would likely be uninformative and limited in value. The authors agree that this is a shortcoming of the paper. However, the purpose of the study was not to fully characterize specific viruses, but to broadly screen for viruses in wildlife (especially bats) living in close proximity to human communities, and thus to better understand potential sources of zoonotic viral transmission in the context of these human-wildlife interfaces. The discussion has been expanded to further discuss this.

• Line 178 Since you used cyt B for species identification: why are the four host species of the pooled guano sampled aren’t identified? The reader might wonder.

Thank you very much for identifying this error! It has come to the attention of the authors that the initial table and figure were created with an older version of the data, without the final host species barcoding incorporated. This has been corrected in the body of the manuscript, in Table 1, and in Figure 1. As such, there are no positive samples where the host species was not identified by barcoding. The four pooled guano samples referred to in Line 178 were attributed to Chaerephon plicatus. 

Reviewer #2: 

• In the Methods section, it is not clear how much of the initial sample volume used for RNA extraction from each of the oral, rectal and guano samples. Although one would expect, this is specially important since the oral swabs yielded the lowest. 

The authors agree with this assessment. The manuscript has been revised to reflect our methodology: samples were collected in 500 μl Trizol, and 250 μl was used from each sample for extraction. One aliquot was used for extraction per the RNA extraction kit, and an additional aliquot was stored to ensure that sufficient volume was available for a second extraction, or other downstream analyses if needed. 8 μl RNA (i.e. the maximum volume for the Superscript III kit) was used for cDNA transcription. 

• Additionally, if deposited in GenBank or elsewhere the sequence data accession numbers need to be provided and mentioned here. 

The authors are in agreement with the reviewer. The sequence data have been uploaded to NCBI GenBank since submission of the first draft of the manuscript, and the accession numbers have been added in a supplemental table. At the time of resubmission, the sequences have not yet been published.

• In the Results and discussion, Authors need to indicate if at all any virus culture been attempted and if any of these viruses has been isolated and characterized.

Thank you for identifying this omission. The authors have corrected the manuscript to reflect that viral culture and isolation were not pursued in this study. 

• The absence of at least RdRp partial sequence phylogenetic analysis is a shortcoming for this report that needs to be discussed or addressed. On a similar note, a phylogenetic analysis derived from spike(S) and receptor-binding domain (RBD) would be informative to include here. Moreover, this report could be further improved if they consider providing information on the protein sequence alignment of coronavirus RdRps or S proteins with the conserved motifs and other unique signatures indicated as necessary. 

Thank you for the suggestions. The authors note that only short fragments of the RdRp gene (328bp and 434 bp, depending on the assay used) were amplified. Given the short sequence fragment of a conserved gene, protein sequence analysis would not be informative. This is the same reason why a phylogenetic analysis was not pursued. The authors agree that this is a shortcoming of the paper. However, the purpose of the study was not to fully characterize specific viruses, but to broadly screen for viruses in wildlife (especially bats) living in close proximity to human communities, and thus to better understand potential sources of zoonotic viral transmission in the context of these human-wildlife interfaces. The discussion has been expanded to further discuss this.

• Although their limited sequencing results may not allow, if the novel viruses found in this study would constitute recent recombination events from existing coronaviruses need to be at least discussed since such recombination is rather common and are thought to contribute to the emergence of novel coronaviruses. 

Thank you very much for the suggestion. However, the authors note that it is not possible to assess recombination from a short fragment of the RdRp gene as was performed in this study; thus, those analyses were not performed and therefore not included in the discussion. We agree that should complete genomes have been amplified, or the complete spike gene sequence amplified, those analyses to identify recombination and associated discussion may have been appropriate. The discussion has been expanded to further discuss this.

---

## [Decision Letter · Decision Letter 1]

10 Mar 2020

Detection of Novel Coronaviruses in Bats in Myanmar

PONE-D-19-34833R1

Dear Dr. Valitutto,

We are pleased to inform you that your manuscript has been judged scientifically suitable for publication and will be formally accepted for publication once it complies with all outstanding technical requirements. In addition, **please amend the '2019-nCoV' and the related nomenclature into SARS-CoV-2 and COVID within the text, whenever appropriate in your finalized version.**

With kind regards,

Renee W.Y. Chan, Ph.D.

Academic Editor

PLOS ONE

Reviewers' comments:

Reviewer's Responses to Questions

**Comments to the Author**

1. If the authors have adequately addressed your comments raised in a previous round of review and you feel that this manuscript is now acceptable for publication, you may indicate that here to bypass the “Comments to the Author” section, enter your conflict of interest statement in the “Confidential to Editor” section, and submit your "Accept" recommendation.

Reviewer #2: All comments have been addressed

2. Is the manuscript technically sound, and do the data support the conclusions?

Reviewer #2: Yes

3. Has the statistical analysis been performed appropriately and rigorously? 

Reviewer #2: Yes

4. Have the authors made all data underlying the findings in their manuscript fully available?

Reviewer #2: Yes

5. Is the manuscript presented in an intelligible fashion and written in standard English?

Reviewer #2: Yes

6. Review Comments to the Author

Reviewer #2: The current revision of this paper has been adequately carried out and it is suitable for publication.

7. PLOS authors have the option to publish the peer review history of their article (what does this mean?). If published, this will include your full peer review and any attached files.

Reviewer #2: No

---

## [Editor Report · Acceptance letter]

26 Mar 2020

PONE-D-19-34833R1 

Detection of novel coronaviruses in bats in Myanmar 

Dear Dr. Valitutto:

I am pleased to inform you that your manuscript has been deemed suitable for publication in PLOS ONE. Congratulations! Your manuscript is now with our production department. 

With kind regards,

on behalf of

Dr. Renee W.Y. Chan 

Academic Editor

PLOS ONE